# Network-Based Driving Force of National Economic Development: A Social Capital Perspective

**DOI:** 10.3390/e23101276

**Published:** 2021-09-29

**Authors:** Lizhi Xing, Xi Ai, Jiaqi Ren, Dawei Wang

**Affiliations:** 1College of Economics & Management, Beijing University of Technology, Beijing 100124, China; aixi@emails.bjut.edu.cn (X.A.); renjq@emails.bjut.edu.cn (J.R.); wangdawei@emails.bjut.edu.cn (D.W.); 2International Business School, Beijing Foreign Studies University, Beijing 100089, China

**Keywords:** global value chain, inter-country input-output table, complex network model, social capital, structural equation model

## Abstract

Network science has been widely applied in theoretical and empirical studies of global value chain (GVC), and many related articles have emerged, forming many more mature and complete analytical frameworks. Among them, the GVC accounting method based on complex network theory is different from the mainstream economics in both research angle and content. In this paper, we build up global industrial value chain network (GIVCN) models based on World Input–Output Database, introduce the theoretical framework of Social Capital, and define the network-based indicators with economic meanings. Second, we follow the econometric framework to analyze the hypothesis and test whether it is true. Finally, we study how the three types of capital constituted by these indicators interact with each other, and discuss their impact on the social capital (economic development level, i.e., GDP). The results prove that the structural capital (industrial status) has a positive impact on the social capital; the relational capital (industrial correlation) has a positive impact on both social capital and structural capital; the cognitive capital (industrial structure) has a small impact on the social capital, structural capital, and relational capital.

## 1. Introduction

Economic development usually spirals upward. In recent years, as the trend of deglobalization has become increasingly prevailing, the patterns of world trade and industrial division of labor have undergone major adjustments, coupled with the widespread and far-reaching impact of the COVID-19, leading to the tremendous shocks faced by the global industrial chain and supply chain. In the complex and volatile international environment, more deeds are to be done by countries to achieve steady GDP growth and bigger relative competitive advantage on the Global Value Chain (GVC). It is necessary not only to optimize domestic industrial layout and improve weak links on the industrial chain and supply chain, but also to give full play to international market resources and their distinctive competitive advantages in international trade. Therefore, it is of significance to study the operating mechanism of the global economic system from the perspective of GVC, so as to enhance the country/region’s relative competitive advantage.

The global economic system is a complex nonlinear system, featuring multiple emergences which will not occur merely through the linear addition of individualities. That is to say, the study of individuals itself may shadow the whole picture. Instead, focus should be put on the interrelationship and influence mechanism between individuals and the whole from the perspective of systems science. All complex systems have their unique topological structures, and their functions often depend on the characteristics of the microstructures. In other words, the prerequisite of understanding the internal mechanism of an economic system is to gauge the structural complexity of the entire system, which, fortunately, is made easier by the constantly developing complex network technology. It is now an important and trendy research topic to model the global economic system based on complex network theory and analyze the topological characteristics and its evolution.

In order to measure the status and function of a country/region on the GVC, and study the causal relations between the industrial layout of an economy and its economic development, we introduce six types of network characteristic indicators and summarize them into the analytical framework of Social Capital, which can be explained in three dimensions, i.e., Cognitive Capital, Relational Capital, and Structural Capital according to the research of Nahapiet and Ghoshal [1], as shown in Figure 1.

The purpose of this paper is to theoretically and empirically enrich the GVC accounting system with the tools from econophysics and econometrics, thus adding up to the existing theoretical framework. It is organized as follows. The related studies are summarized in Section 2. Section 3 presents the data, model, and setups. Section 4 builds up the analytical framework from a network-based social capital perspective. Section 5 carries out the hypothesis testing, which is followed by the discussions of causal relationship among and between dependent and independent variables in Section 6. Finally, conclusion is provided in Section 7.

## 2. Literature Review

Social capital originated from the concept of capital in economics. With the deepening of research, scholars in different fields such as sociology and political science have defined social capital from diverse perspectives [2]. Bourdieu was the first scholar to clearly put forward the concept of social capital. From the perspective of social networks, Bourdieu and Wacquant believed that social capital is the sum of resources accumulated by a network composed of individuals or their relationships. Different from the study of social capital theory at the individual level, Burt first extended the theory of social capital to the enterprise level, thinking that social capital is a kind of resource that an enterprise obtains from a social network as a purposeful social actor [3]. Burt’s famous Structure Hole Theory emphasizes the importance of entrepreneurs occupying a favorable brokerage position in the relationship network to provide resources for enterprises. More and more scholars pay their attention to the macro-level of social capital, which is to regard it as the resources and wealth possessed by an organization, a community, or even the entire society. Putnam believed that social capital is an organizational feature, such as trust and norms. The economic and democratic development of a society is largely determined by the extent of richness of its social capital [4].

Social capital theory and social network theory are inseparable. Initially, social network theorists represented by Coleman, Lin, and Burt constructed the concept of social capital at the micro and meso analysis levels within the framework of social network theory. Yet the impact of macro-level social capital needs to be further studied [5]. In their in-depth analyses, scholars divided social capital into multiple dimensions accordingly. Coleman emphasized the structural attributes of social capital, and believes that social capital is a social “structural resource” that is determined by its function [6]. Putnam divided social capital into two dimensions: bridging social capital and bonding social capital [4]. Nahapiet and Ghoshal adopted a resource-based organization view to illustrate the relationship between social capital development and organizational performance [7], arguing that social capital has structural, relational, and cognitive dimensions [1]. In this paper, we explore the internal relations between social capital and its three dimensions represented as the industrial status, industrial correlation, and industrial structure on the GVC.

## 3. Data

The Inter-Country Input-Output (ICIO) data are utilized in this paper not only for its ability of reshowing flows of intermediate products, final goods, and services but also for possible comparison on the same basis. Given that a world economy with m countries/regions (u,v=1,2,…,m), n sectors within each country/region, and totally N=m×n sectors (i,j=1,2,…,N), as shown in Table 1 (referring to our former study [8]).

In the ICIO table, $Z^{uv}$ is an n×n matrix of intermediate inputs that are produced in country u and used in country v; $Y^{uv}$ is a n×1 vector giving final products produced in country u and consumed in country v; $X^{u}$ is also an n×1 vector giving gross outputs in country u; and $VA^{u}$ denotes a n×1 vector of direct value-added in country u [9,10]. To depict the transmission of value stream on the GVC, we take the region of inter-country inter-industry use and supply as the modeling data source, i.e., the $Z^{uv}$ matrix, in which row vectors record the allocation of outputs and column vectors, the composition of demand.

The data of this paper are derived from the World Input-Output Database (WIOD) released in 2016 (WIOD2016 for short), which is value-type Input-Output (IO) table and covers the trade data of intermediate and final goods of 56 industrial sectors in 43 countries/regions and Rest of World (RoW) from 2000 to 2014 [11]. Data for 56 sectors are classified according to the International Standard Industrial Classification Revision 4. In detail, the names and abbreviations of countries are listed in Table A1, and those of industrial sectors are listed in Table A2 in the part of Appendix B.

The part of intermediate use in WIOD2016 is straightforwardly adopted to build graph G=(V,E,W), containing N nodes, drawn to represent sectors within a country/region and denote a node set V. Pairs of nodes are linked by edges to reflect their interdependencies, thereby forming an asymmetric edge set E. In valued graphs, a set E can be replaced by weight set W, which is just the region of inter-country and inter-industry use and supply in ICIO table. Finally, we name this weighted and directed network the Global Industrial Value Chain Network (GIVCN) model.

## 4. Methodology

The issue studied in this paper is based on the theoretical framework of social capital, and we use network-based indicators to measure the impact of structural capital, relational capital, and cognitive capital on the country/region’s macroeconomic performance.

### 4.1. Structural Capital: Industrial Status

Structural capital is mainly used to characterize the status of industrial sectors on the GVC, i.e., the macro performance of the industrial sector and its participation in worldwide synergic production. According to our previous study [12], the Random Walk Centrality (denoted as $C_{RC}$) can measure the national competitiveness on the GVC. The derivation process of GIIC is shown in Figure 2 and the details are attached in the Section B.1 Supplementary Description of GIIC.

In order to measure the relative competitive advantage of a country/region and analyze the interrelation between national industrial impact and its global economic status, we define the Global Industrial Impact Coefficient (GIIC).

GIIC is hence derived from the sum of $C_{RC}$ of all the sectors within each country/region to evaluate the national competitiveness on the GVC:(1)$$GIIC\left(u\right)=10^{3}\times \underset{i\in \tau \left(u\right)}{\sum }C_{RC}\left(i\right)$$
where GIIC(u) is the GIIC of country/region u; τ(u) is a set of numbers standing for the row sequence number of a certain country/region in the adjacent matrix $Z^{uv}$. For instance, China is the 8th economy in WIOD2016, so τ(8)={393,394,⋯,448} because each economy owns n=56 sectors. Besides, due to the huge size of the GIVCN model, the value of $C_{RC}$ is rather small, so the sum is timed by one thousand for convenience.

In addition, the GVC well depicted by ICIO data is a sort of complex network from the perspective of econophysics, and we can further investigate the relationship between final demand and intermediate goods production. According to our previous study [8], the Counting First Passage Betweenness (denoted by $C_{FP}$) tracks how often a given node is visited on the first-passage walks between all source-target pairs. The derivation process of GDDI is shown in Figure 3 and details are attached in the Section B.2 Supplementary Description of GDDI.

In order to measure the country/region-sector’s participation in worldwide synergic production, we define the Global Demand Dependence Index (GDDI).

In the GIVCN model, therefore, the sum of $C_{FP}$ of all the sectors within each country/region is defined as the GDDI. Compared with previous studies, this method can better capture the instantaneous dynamic characteristics of value flow.
(2)$$GDDI\left(u\right)=\underset{i\in \tau \left(u\right)}{\sum }C_{FP}\left(i\right)$$
where GDDI(u) is the GDDI of country/region u. GDDI is here adopted to measure country-sector’s participation in worldwide synergic production, i.e., the bigger the sector’s GDDI the higher the degree of globalization.

### 4.2. Relational Capital: Industrial Correlation

The relational capital in this article is mainly used to characterize the industrial correlation, i.e., to measure the industrial sectors’ interdependence with all the upstream and downstream ones. The relative position of the industrial sector on the GVC could be reflected by Backward Closeness and Forward Closeness of Industrial Sectors, and the comparative advantage brought by different locations. Specifically, a single sector on the supply-side is called the upstream sector once it directly or indirectly provides intermediate products or services to one consumer at least, while a sector on the demand-side is taken as the downstream sector only if it directly or indirectly consumes intermediate products or services, even from the sole provider. Accordingly, we calculate $C_{c}^{RFWA-IN}$ and $C_{c}^{RFWA-OUT}$ based on the numerical matrix of Strongest Relevance Path Length (SRPL) [13], which are the counterparts of in-degree and out-degree closeness centrality in complex networks. According to our previous study, the derivation process of $C_{c}^{RFWA-IN}$ and $C_{c}^{RFWA-OUT}$ is shown in Figure 4 and details are attached in the Section B.3 Supplementary Description of Backward Closeness and Forward Closeness.

In order to quantify the closeness of relation among countries/regions, we design two indicators at the national level based on the $C_{c}^{RFWA-IN}$ and $C_{c}^{RFWA-OUT}$, named the National Industrial Backward Closeness (NIBC) and National Industrial Forward Closeness (NIFC):(3)$$NIBC\left(u\right)=\underset{i\in \tau \left(u\right)}{\sum }C_{c}^{RFWA-IN}\left(i\right)$$
(4)$$NIFC\left(u\right)=\underset{i\in \tau \left(u\right)}{\sum }C_{c}^{RFWA-OUT}\left(i\right)$$
where NIBC(u) and NIFC(u) are the backward closeness and forward closeness of country/region u, respectively.

### 4.3. Cognitive Capital: Industrial Stability

Different from structural capital and relational capital, the two indicators of cognitive capital are derived from Global Industrial Value Chain Network Bipartite Graph-Filtering Edges (GIVCNBG-FE) model [14]. Our modelling framework is shown in Figure 5.

The cognitive capital is mainly used to characterize the degree of perfection of the industrial structure by measuring the degree of nested structure of the network. Higher degree of nestedness of GVC network indicates the stable industrial structure, the mature industrial trade mechanism, the regular and orderly industrial trade network, and the deeper integration between industries [15]. The Nested Overlap and Decreasing Fill (NODF) metric proposed by Almeida-Neto et al. is generally applied to calculate the nestedness of the network, which is based on two basic properties: Decreasing Fill (DF) and Paired Overlap (PO) [16] (see the Section B.4 Supplementary Description of NODF for details.)

Considering that a country/region’s macroeconomic performance is affected by both domestic and international trade cycles, we define two indicators, i.e., Export Trade Network-Nested Overlap and Decreasing Fill (ETN-NODF) and Import Trade Network-Nested Overlap and Decreasing Fill (ITN-NODF), to measure the nestedness of the local network in terms of economies.
(5)ETN−NODF(u)=NODF(Export Trade Network of Country/Region u)
(6)ITN−NODF(u)=NODF(Import Trade Network of Country/Region u)
where, ETN−NODF(u) measures the nestedness of export trade network of country/region u, which is formed while industrial sectors within country/region u, as upstream sectors (in the supply side), export intermediate goods to the others; in the opposite, ITN−NODF(u) measures the nestedness of the import trade network, which is formed while industrial sectors within country/region *u*, as downstream sectors (in the demand side), import intermediate goods from the others.

### 4.4. Econometric Model

After Structural Equation Model (SEM) had been used to analyze the causality between latent variables [17], Wold created Partial Least Squares (PLS) as a complementary approach to factor-based SEM [18]. As a popular research tool, PLS can test hypotheses in an exploratory way, especially in complex path models with relaxed expectations on data [19]. In recent years, PLS-SEM has become popular in management, social sciences, and psychology [20]. According to Ringel and Sarstedt, PLS-SEM is a path model for estimating latent variables based on variance and is especially useful in key interpretation sources of a target structure [21], and in identifying relationships between constructs [22]. However, it is easy to ignore the mediating effect that does not directly influence the complex path models. Nitzl et al. provided decision tree and high-level mediating effect classification, which helps improve the accuracy [20]. Hair et al. used the Finite Mixture PLS (FIMIX-PLS) module of SmartPLS 3 software [23] based on a popular corporate reputation model, identified and processed the unobserved heterogeneity in PLS-SEM [24,25]. In addition, to evaluate the reliability and validity of higher-order concepts in applied social science research, Sarstedt et al. used the well-known corporate reputation model to prove and estimate the reflective-reflective and reflective-formative types of higher-order constructs [26].

Since it offers the flexibility needed for the interplay between theory and data [27], PLS-SEM is becoming more and more popular in modelling the causality [28]. For instance, it is usually applied to analyze the secondary data, such as social media data, national statistical bureaus, or publicly available survey data. Richter et al. combined PLS-SEM and Necessary Condition Analysis (NCA) as complementary views of causality and data analysis [29]. Khan et al. applied Social Network Analysis (SNA) to investigate the knowledge network structure of PLS-SEM and identified the key journals for network knowledge dissemination [30]. The relevant theoretical results of PLS-SEM have fully proved that it has great advantages in causal inference. Hence, combined with the social capital theory, we use the PLS-SEM model and SmartPLS3 software to explore the relationships between various types of capital and the level of national economic development.

## 5. Hypotheses

### 5.1. Hypothesis Formulation

First, GIIC and GDDI (as structural capital) can effectively reflect the global industrial influence and the participation in worldwide synergic production. The GIIC as a part of structural capital, can fully measure one country/region’s competitiveness in gaining information superiority and intermediate interests. A higher GIIC often implies a greater GDP level. On the other hand, GDDI reflects the cumulative effect of global market demands when directly and indirectly relevant sectors are involved in the global production system. The bigger a country/region’s GDDI, the higher its degree of globalization. Accordingly, we propose the hypothesis:

**Hypothesis** **1.** *Structural capital should positively affect social capital*.

Next, NIBC and NIFC (as relational capital) can effectively reflect the impact of industrial correlation on the macroeconomic and industrial status, the greater $C_{c}^{RFWA-IN}$ (Backward Closeness) and $C_{c}^{RFWA-OUT}$ (Forward Closeness), the more it can reflect the closeness of industrial correlation. In other words, if the NIBC and NIFC become greater, there will be stronger compactness between certain country/region and its upstream or downstream counterparts. It urges economies to reinforce the ability to integrate upstream or downstream industrial resources. A country/region with a stronger industrial correlation is more conducive to occupying a dominant industrial status, which is inseparable from the level of national economic development. Similarly, the closer the industrial correlation (relational capital), the higher the level of economic development (social capital) and industrial status (structural capital). From this, we put forward the following hypotheses:

**Hypothesis** **2.** 
*Relational capital should positively affect social capital.*


**Hypothesis** **3.** 
*Relational capital should positively affect structural capital.*


Finally, ETN-NODF and ITN-NODF (as cognitive capital) can measure the nestedness of the local network in terms of economies. A country/region with better macroeconomic performance generally has higher degree of nestedness, indicating that their trade mechanisms are relatively mature, the global industrial layout is reasonable, and the industrial structure is nearly complete. For the effects of cognitive capital on social capital, the more perfect a country/region’s international trade network operating mechanism, the stronger its economic vitality. In other words, the stability of the export and import structures and the coordinated development of industrial sectors will help develop national economy. For the effects of cognitive capital on structural capital, the completeness of the industrial structure within a country/region determines its relative competitiveness and influence on the GVC. In addition, regarding the role of cognitive capital in promoting relational capital, a complete and stable industrial structure is also an indispensable prerequisite for close correlation and coordinated development between industries. Thus, the degree of industrial structure perfection (cognitive capital) has a certain impact on the level of macroeconomic development (social capital), industrial status (structural capital), and industrial correlation (relational capital). The more stable the industrial structure, the higher the level of macroeconomic development; the higher the industrial status, the stronger the industrial correlation. Accordingly, we propose the following hypotheses:

**Hypothesis** **4.** 
*Cognitive capital should positively affect social capital.*


**Hypothesis** **5.** 
*Cognitive capital should positively affect structural capital.*


**Hypothesis** **6.** 
*Cognitive capital should positively affect relational capital.*


Figure 6 shows the hypothesis conceptual framework.

### 5.2. Hypothesis Testing

The model calculation data as Appendix A named "PLS-SEM data.xlsx". In order to reduce data heterogeneity, we take the logarithm of all indicators. Descriptive statistics and the correlation matrix of estimated variables are illustrated in Table 2.

According to the research of Hair, Ringle, and Sarstedt [31], PLS approach is applicable to evaluate the influence of social capital theory on macroeconomic development [32]. Strictly speaking, the desired level of the ratio is between 15 and 20 observations for each independent variable [33]. However, there are 630 observations and six independent variables in this study, leaving little concern of small-sample bias. To examine the specific effect of each indicator, we first conduct path analysis for all variables as shown in Figure 7.

Overall, structural capital positively affects social capital (H1: β = 0.604, *p* < 0.001), so H1 is supported. Relational capital positively affects social capital and structural capital (H2: β = 0.382, *p* < 0.001; H3: β = 0.941, *p* < 0.001), so H2 and H3 are supported. Cognitive capital positively affects social capital, structural capital, and relational capital (H4: β = 0.024, *p* < 0.001; H5: β = 0.036, *p* < 0.01; H6: β = 0.426, *p* < 0.001), so H4, H5, H6 are supported.

Through the bootstrapping operation in the SmartPLS3 software, the data shown in Table 2 are obtained. T-statistics are all greater than 1.96, and *p*-values are all less than 0.05. Therefore, all the three capitals have significant positive effect on social capital.

In order to assess the effect size of a particular independent variable on a dependent variable [31], the squared multiple (or multiple partial) correlation ($R^{2}$) is used to calculate Cohen’s $f^{2}$ [34]. The formula of effect size is as follows.
(7)$${\mathrm{Cohen}}^{\text{’}}s\;f^{2}=\frac{R_{full}^{2}-R_{reduced}^{2}}{1-R_{full}^{2}}$$
where $R_{full}^{2}$ is the value of $R^{2}$ from the least-square model that includes all independent variables; $R_{reduced}^{2}$ is the value from that includes all but one particular set of independent variables [32,34].

As Cohen proposed in his research [33], an effect size of 0.02 ≤ $f^{2}$ < 0.15 is small; 0.15 ≤ $f^{2}$ < 0.35 is medium; and $f^{2}$ ≥ 0.35 is large. Table 3 also shows the effect sizes of all significant hypothesized associations. It is confirmed that industrial status has a large effect size ($f^{2}=1.142$) on the level of economic development (GDP). For the industrial correlation, it has a large effect size both on GDP ($f^{2}=0.464$) and industrial status ($f^{2}=8.545$). However, for the industrial structure, it has a small effect size on the GDP ($f^{2}=0.017$) and industrial status ($f^{2}=0.013$). Likewise, industrial structure has a medium effect size ($f^{2}=0.221$) on the industrial correlation.

## 6. Results and Discussions

This study regards the level of economic development as social capital, and for the first time uses the three dimensions of social capital theory to examine the impact of each dimension on the level of national economic development. The effects from each dimension are discussed below.

### 6.1. The Effects of Structural Capital

From the results of PLS-SEM model, structural capital (industrial status) positively affects social capital (GDP) with a large beta coefficient and a large effect size ($f^{2}=1.142$). The positive correlation between GIIC and GDP indicates that one country/region’s global industrial impact can reveal its international competitive advantage. The stronger the industrial influence, the greater the economic strength and the broader prospects for development. In addition, the higher the degree of participation in worldwide synergic production (GDDI), the higher the degree of globalization. Specifically, focusing on industrial sustainability, expanding industrial influence, and increasing participation in worldwide synergic production can enable a country/region to occupy a favorable industrial status on the GVC. This can not only contribute to the GVC, but also stimulate the macroeconomic growth. In other words, strengthening the accumulation of structural capital can promote the development of social capital.

As the world’s largest economy, the United States provides developing countries/regions with high-value-added intermediate goods, and integrates domestic value chains to enhance its internationalization. It is precisely because of the dominant position of the United States and the high connectivity of economic activities that the subprime mortgage crisis quickly spread to other economies worldwide and trigger the global economic tsunami. This fully indicates that in the process of global integration, once a crisis occurs in a country/region with high global industrial influence and high industrial status, the global economic system will be severely affected.

### 6.2. The Effects of Relational Capital

Relational capital composed of NIBC and NIFC well explains the internal mechanism forming the competitiveness of country/region. The greater the both, the stronger the ability of the economy to connect with the supply side or the demand side, and the more prominent its competitive advantage on the GVC. We further optimize the model and find that NIBC is more applicable to measure relational capital. This is because that the higher the NIBC, the closer it is to a relatively downstream position on the GVC; the stronger its ability to create added value, the more intermediate goods it can provide for downstream consumers. In other words, being in a relatively downstream position means that the economies are closer to the market. According to the Smile Curve Theory, the relative competitive advantage would be more prominent if owning more market resources. Thus, we can identify the relative position of an economy based on relational capital and analyze its preferable types of services and industrial status.

On the one hand, relational capital (industrial correlation) positively affects social capital (GDP) with a large beta coefficient and a large effect size ($f^{2}=0.464$). This also reflects that the closer the relationship between the economy and its upstream or downstream counterparts, the richer the market resources it obtains, and the higher the degree of industrial correlation. As for countries/regions, industrial correlation is the basis of sustainable industrial development. The positive correlation between relational capital and social capital indicates that the stronger the degree of industrial correlation, the higher the level of national macroeconomic development.

On the other hand, relational capital (industrial correlation) is positively correlated with structural capital (industrial status) with a large beta coefficient and a large effect size ($f^{2}=8.545$). In addition, an economy with more market resources will be more closely connected with its upstream and downstream counterparts, which means that the stronger the ability to integrate resources, the more conducive to the formation of complete Industrial Value Chain (IVC) network. In the meanwhile, this type of country/region often plays a role in linking pieces of GVC, which bring itself a higher industrial status on the GVC in turn. Specifically, relational capital and structural capital are positively correlated, i.e., the industrial sector with stronger industrial correlation always has a higher industrial status.

The recent trend of economic globalization is the formation of Regional Value Chain (RVC). For instance, the “European Factory” centered on Germany is one of the three cross-border production systems, in which more and more production factors could circulate on the European RVC [35]. Germany imports industrial raw materials and intermediate products from other European Union members, and then exports the reprocessed and manufactured products to them or other countries/regions around the world. From this angle, Germany plays the dual role of the European trade center and the GVC hub between Europe and the world. The case of German manufacturing industry tells us that, strengthening industrial correlation and focusing on the accumulation of relational capital can promote the development of structural capital and social capital.

### 6.3. The Effects of Cognitive Capital

First, cognitive capital (industrial structure) positively affects social capital (GDP) with a small beta coefficient and a small effect size ($f^{2}=0.017$). The cognitive capital incorporating ETN-NODF and ITN-NODF represents the completeness of industrial structure. If a country/region’s ETN-NODF is higher, it will act as a supplier trading intermediate goods with other countries, thus forming a relatively nested export trade network. In the opposite, if the ITN-NODF is higher, there will be a relatively nested import trade network. In our opinion, the nested structure stands for the maturity of trade cooperation mechanism. Thus, the process of continuously optimizing the industrial structure is that of the accumulation of cognitive capital, which will benefit to the economic development at the macro level.

Second, cognitive capital (industrial structure) positively affects structural capital (industrial status) with a small beta coefficient and a small effect size ($f^{2}=0.013$). Countries/regions with a nested RVC or GVC network can gain greater participation in global resource allocation and acquire advantageous position in the international division of labor. In sum, cognitive capital needs to be continuously improved in accumulation to achieve the goal of promoting structural capital.

Third, cognitive capital (industrial structure) positively affects relational capital (industrial correlation) with a large beta coefficient and a medium effect size ($f^{2}=0.221$). The rationality and integrity of the industrial layout enable products and values to flow effectively on the IVC, resulting in the close industrial correlation and the highly interdependent trade relationship. To achieve the goal of promoting national economic development, a rational distribution of foreign trade (cognitive capital) is necessary to further consolidate and enhance industrial status and competitive advantage on the GVC.

With its accession to the World Trade Organization (WTO), China has been actively participating in the international division of labor at different levels, maintaining a good momentum of development. China boasts the world’s most complete industrial system and steady industrial support ability, both ensure its strong resilience of economy. After the subprime mortgage crisis, China adjusted its industrial layout through industrial transformation and upgrading, to better participate in resource allocation and market competition on a global scale. As a result, its competitiveness of new technology-intensive industries has been greatly enhanced. In addition, China’s huge domestic consumer market and potentials have also accelerated the accumulation of cognitive capital, leading to the continuously optimized and upgraded industrial structure. However, the COVID-19 pandemic has exposed the shortcomings of China in the fields of high-end equipments and products, primary agricultural product, and important mineral resources. This will inevitably push China to form a more comprehensive industrial layout that takes into account both domestic and foreign markets. In addition, China’s implementation of Supply-Side Structural Reforms and Dual-Circulation Strategy urges itself to coupling with the new dynamics of deglobalization of the world economy.

## 7. Conclusions

In this paper, an industrial complex network is built up according to the basic framework of econophysics and an econometric analysis framework is adopted to test the relevant hypotheses. It explains why social capital can be used to analyze the causality between microstructure and macro performance in the GVC network. In details, we construct the GIVCN-WIOD2016 model, define six types of indicators with economic meanings based on the three dimensions of social capital, and explain how they interact and affect the development of the national economy. More importantly, we provide an analytical framework to summarize the driving factors of national economic development in the context of globalization, and taking the main economies in the world as examples. By doing so, we have laid the foundation for further theoretical development and empirical research.

## Figures and Tables

**Figure 1 entropy-23-01276-f001:**
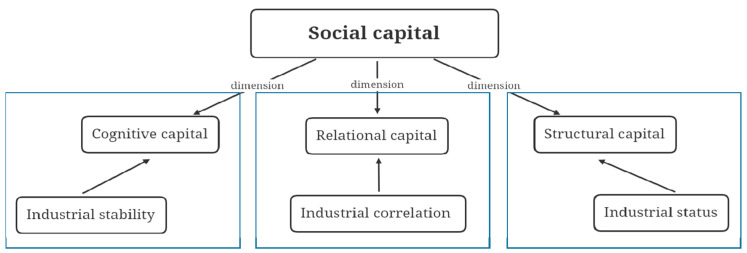
Conceptual map.

**Figure 2 entropy-23-01276-f002:**
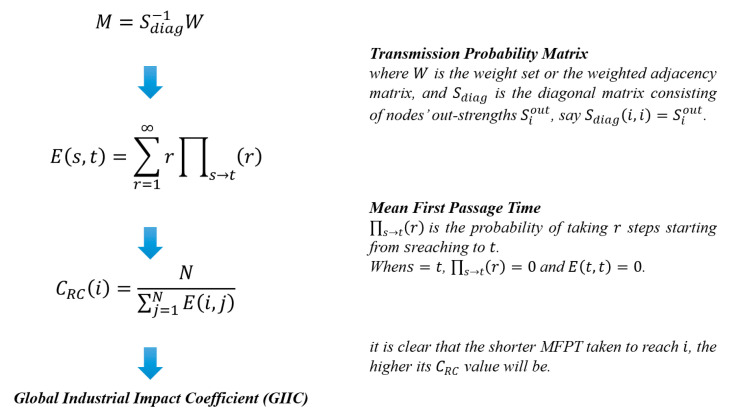
Formula derivation of GIIC.

**Figure 3 entropy-23-01276-f003:**
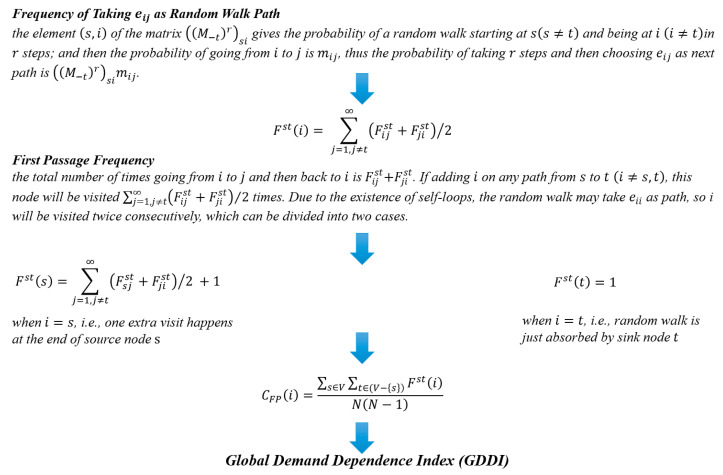
Formula derivation of GDDI.

**Figure 4 entropy-23-01276-f004:**
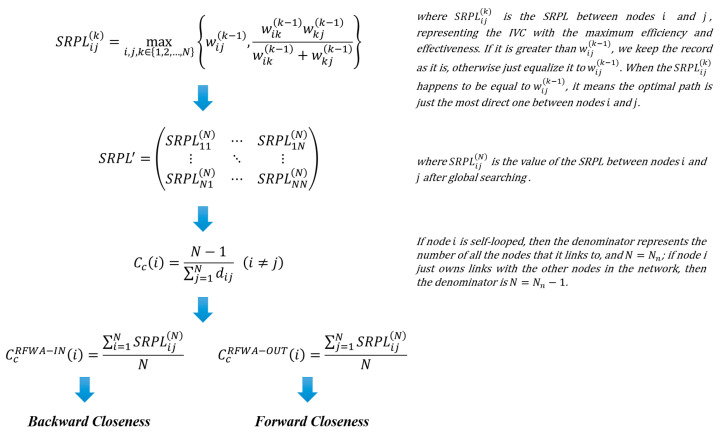
Formula derivation of backward closeness and forward closeness.

**Figure 5 entropy-23-01276-f005:**
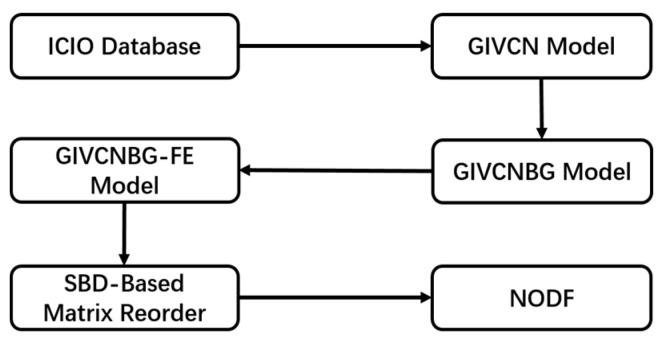
Computational process of NODF.

**Figure 6 entropy-23-01276-f006:**
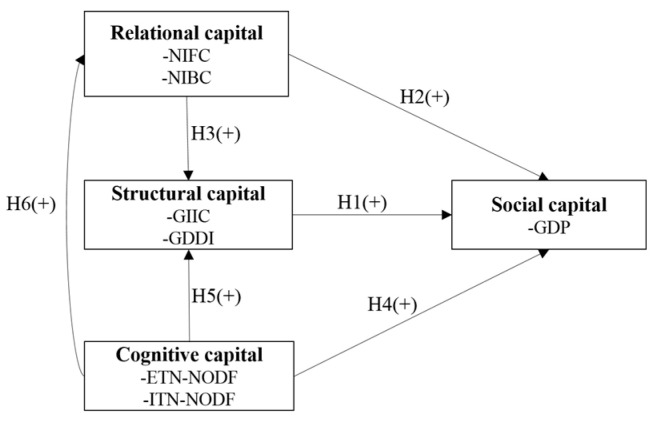
The hypothesis conceptual framework.

**Figure 7 entropy-23-01276-f007:**
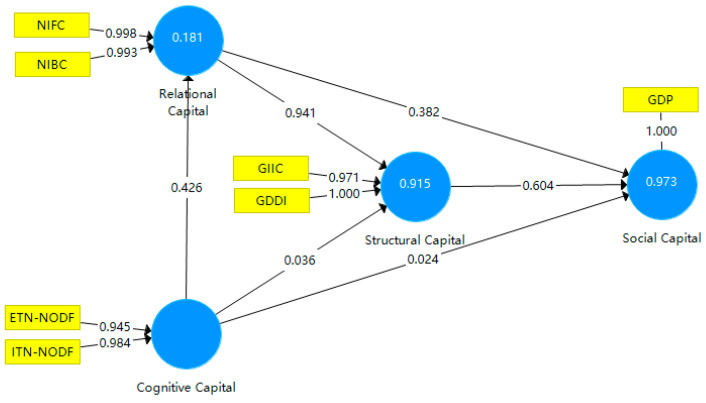
Tests for the hypothesized associations.

**Table 1 entropy-23-01276-t001:** The layout of ICIO table.

		Output		Intermediate Use			Final Demand			Total Output

Input		Country		Country *A*	Country *B*	ROW	Country *A*	Country *B*	ROW	
			Sector	*A*_1_, …, *A*_n_	*B*_1_, …, *B*_n_	*R*_1_, …, *R*_n_	*A*_1_, …, *A*_n_	*B*_1_, …, *B*_n_	*R*_1_, …, *R*_n_	
Intermediate Inputs		Country *A*	*A*_1_, …, *A**_n_*	*Z^AA^*	*Z^AB^*	*Z^AR^*	*Y^AA^*	*Y^AB^*	*Y^AR^*	*X^A^*
		Country B	*B*_1_, …, *B_n_*	*Z^BA^*	*Z^BB^*	*Z^BR^*	*Y^BA^*	*Y^BB^*	*Y^BR^*	*X^B^*
		ROW	*R*_1_, …, *R_n_*	*Z^RA^*	*Z^RB^*	*Z^RR^*	*Y^RA^*	*Y^RB^*	*Y^RR^*	*X^R^*
		Value-Added		*VA^A^*	*VA^B^*	*VA^R^*	**Notes:**$Z^{uv}=\left(\begin{array}{ccc} Z_{u_{1}v_{1}} & \cdots & Z_{u_{1}v_{n}}\\ \vdots & \ddots & \vdots \\ Z_{u_{n}v_{1}} & \cdots & Z_{u_{n}v_{n}} \end{array}\right)$;$X^{A}=VA^{A}+Z^{AA}+Z^{BA}+\ldots +Z^{RA}$;$X^{A}=Z^{AA}+Z^{AB}+\ldots +Z^{AR}+Y^{AA}+Y^{AB}+\ldots +Y^{AR}$.			
		Total Input		*X^A^*	*X^B^*	*X^R^*				

**Table 2 entropy-23-01276-t002:** Descriptive statistics and correlation matrix.

ID		Mean	S.D.	1	2	3	4	5	6
1	GDP	11.497	0.768	1.000					
2	GDDI	3.706	0.773	0.966	1.000				
3	GIIC	1.615	0.714	0.946	0.969	1.000			
4	NIFC	3.698	0.491	0.959	0.952	0.912	1.000		
5	NIBC	3.710	0.470	0.957	0.956	0.910	0.985	1.000	
6	ETN-NODF	0.815	0.384	0.418	0.449	0.447	0.449	0.421	1.000
7	ITN-NODF	0.435	0.395	0.397	0.476	0.463	0.446	0.415	0.856

**Table 3 entropy-23-01276-t003:** Effect sizes of significant hypothesized associations.

Hypothesis	β ^a^	T-Statistic	*p*-Values	$f^{2}$	Effect Size ^b^	Decision
H1: Structural capital→Social capital	0.604	30.028	0.000 ***	1.142	large	Supported
H2: Relational capital→Social capital	0.382	18.560	0.000 ***	0.464	large	Supported
H3: Relational capital→Structural capital	0.941	142.094	0.000 ***	8.545	large	Supported
H4: Cognitive capital→Social capital	0.024	3.525	0.000 ***	0.017	small	Supported
H5: Cognitive capital→Structural capital	0.036	2.764	0.006 **	0.013	small	Supported
H6: Cognitive capital→Relational capital	0.426	14.899	0.000 ***	0.221	medium	Supported

^a^ Type II error in hypothesis testing in statistics. Compare the absolute effect or contribution of each coefficient. ^b^ The overall effect sizes $f^{2}\;$< 0.02, 0.15, or 0.35 are regarded as small, medium, and large effects, respectively. * *p* < 0.05 (2-tailed), ** *p* < 0.01 (2-tailed), *** *p* < 0.001 (2-tailed).

## Data Availability

The data presented in this study are available in “PLS-SEM data.xlsx”.

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
