# Peer review of "Network-Based Driving Force of National Economic Development: A Social Capital Perspective"

_entropy, 2021, doi:10.3390/e23101276_

Round 1

Reviewer 1 Report

I started reading the paper with interest, due to the use of networks and systemic approach to the study of economic development. What I effectively found is not any real innovative view in the analysis of global economies. On the contrary, I found only the confirmation that the evaluation of the strength of an economy must be based on social capital, which is considered by the authors a measure of the GDP. Hopefully, economists will soon replace GDP by more appropriate indicators, able to assess welfare and well being of citizens, as well as environmental preservation. I invite the authors to be aware of the limitations of GDP, that are well reported in literature (https://hbr.org/2019/10/gdp-is-not-a-measure-of-human-well-being). 

Said that, I also find the presentation of the methodology used by the authors poor from the point of view of the statistical indicators. It seems that they have used some software tools as black boxes without giving any explanation of the underlying mathematics. Indeed, while they explain almost clearly the way the data have been arranged to estimate the relationships among sectors and countries, I don't find any explanation of the parameters "beta" and "f^2", as an example. 

I don't find the paper enough interesting for the readers of Entropy and I have some ethical concerns with reducing advanced and modern approaches, such as complex networks, which embed nonlinearities and emergence of unexpected phenomena, as an alternative way to evaluate the social capital (GDP) in the development of sustainable economies.

Reviewer 2 Report

Authors of a manuscript "Network-based driving force of national economic development: a social capital perspective" analyze the Inter-Country Input-Output table data by calculating several indicators related to different aspects of social capital in order to test a few hypotheses on directional influence of these aspects on the overal value of the social capital (here proxied by GDP). The outcomes and conclusions reported in the paper are interesting and deserving publication, but this can be done only after Authors significantly rewrite some parts of the text.

While the introduction, literature review, results, discussion, and conclusion sections are clearly presented, the data, methodology, and hypotheses sections are too sketchy, chaotic, and lack self-containedness. The division into "Data and Model", "Hypotheses", and "Methodology" is not consistent with the actual content of the related sections. I would suggest Authors to rearrange these sections into "Data", "Methodology", and "Hypotheses" (in that order), to leave the content of the present section 3 unchanged (under the title of "Data"), move all the definitions and mathematics from the present sections 4 and 5 into the "Methodology" section (no. 4), and move all the hypotheses to the "Hypotheses" section (no. 5). In the latter, the hypotheses should not only be formulated, but also a rationale behind each of them should be given (for instance, this can be done by moving some content from present section 6).

A lack of self-containedness means that quantities and methods are given or mentioned in the text without their satisfactory description. Authors direct a reader to the literature, but it makes the manuscript hard to follow sometimes, especially when there are quantities introduced without even naming them (C_FP, C_RC) or without providing their interpretation (M, F_st, etc.). The same refers to a very sketchy descriptions given in Figs. 2-4. While such an attitude would be acceptable in a more technically-oriented econometric journal, "Entropy" is in fact addressed to a broader audience and this requires more a detailed presentation. Moreover, Authors repeatedly use confusing index markings. For example, if they assume that the indices r and s denote countries, and the indices i and j denote industrial sectors (section 3), they should not use the same index i to denote countries (Fig. 3 and Eqs.1 and 2). I am afraid that Authors too closely follow Refs.[11-13] by copying formulas given there and add their own content without arranging these things logically correct. Please, be more careful in presenting the mathematical content.

All the acronyms should be extended at their first occurrence. This is not the case in the present manuscript version, where some acronyms are explained much later than they occur and some are not explained at all.

Fig. 1 contains rather a strange attribution of names to dimensions. An idea behind labelling the three dimensions as "points", "lines", and "areas" is obscure.

Round 2

Reviewer 1 Report

The authors have answered satisfactorily to the reviewer's comments. The paper can be accepted for the publication in Entropy.

Author Response

Thank you very much for your comments on our manuscript.